# Microstructure and Properties of Phosphorus Bronze/Brass Joints Produced by Resistance Projection Welding

Ruilin Lai [1,2], Weijun Zhang [2], Xiaofei Sheng [3], Xianjue Ye [1], Yingfeng Cai [2], Xiwei Zhang [2], Temiao Luo [2], Pinghu Chen [4], Qian Lei [1] and Yunping Li [1,*]

1. Powder Metallurgy Research Institute, Central South University, Changsha 410083, China; 133701033@csu.edu.cn (R.L.); leiqian@csu.edu.cn (Q.L.)
2. Gongniu Group Co., Ltd., Ningbo 315314, China; luotm@gongniu.cn (T.L.)
3. School of Mechanical and Electrical Engineering, Wuhan Business University, Wuhan 430058, China
4. School of Mechanical Engineering, University of South China, Hengyang 421200, China; chenpinghu1986@163.com
* Correspondence: lyping@csu.edu.cn

**Abstract:** In this work, we fabricated lap joints between embossed projection phosphorus bronze and flat brass through resistance projection welding (RPW). The experimental results indicated that the bronze projection moves into the softer brass without being deformed during the welding process. The tensile shear loads of the joint reached a maximum value of 273.6 N at a welding current of 5.5 kA. Under this circumstance, a reaction layer, including a columnar crystal solidification layer and a diffusion layer, is formed at the interface beside the boundary of bronze. The EDS line scan shows an elemental transition diffusion layer of about 1.5 μm between the H62 brass columnar crystal and XYK-6 phosphorus bronze. The fracture occurred on the XYK-6 side, passing through the bump instead of the welding interface, resulting in intactness of the welding interface. The results revealed that resistance projection welding is an effective method for welding copper alloys, suggesting the bright prospects of this technology in welding electrical parts.

**Keywords:** resistance projection welding; dissimilar joint; interface; properties; microstructure

## 1. Introduction

Copper alloys, such as bronze and brass, are widely used in the field of electrical conductivity due to their excellent mechanical properties, high electrical conductivity, and high thermal conductivity [1,2]. However, copper alloys are difficult to weld, and they are prone to generate hot cracks and pores, resulting in a significant decrease in the effectiveness of their mechanical properties and electrical conductivity [3,4]. The welding methods of copper alloys mainly include fusion welding [3,4], pressure welding [5,6], and brazing [7,8]. At present, the electrical connection between the copper bars and the modules is made by brazing [9]. However, brazing makes it easy to produce false welding, leading to increased electrical resistance and decreased strength of the welding joint, endangering the safety of electricity consumption. In addition, the production process of brazing generates a lot of smoke and dust, polluting the environment and endangering health. Therefore, it is necessary to study an efficient welding method for copper alloys instead of the brazing method.

Resistance projection welding (RPW) is a type of resistance welding, which has become the most widely used method for welding sheet metals due to various advantages including easy operation, excellent adaptability for automation, high efficiency, and sound quality production [10]. The utilization of projection design can be highly effective in concentrating the welding current and heat generation. Therefore, it is perfectly applied to copper and copper alloys resistance welding, due to the generally high conductivity of these materials [11]. Recently, many studies on RPW have been reported, including

its physical mechanism [10–12], parameters optimization [13–22], microstructure characterization [20–25], and numerical process simulation [16,26]. Wehle et al. [10] revealed that projection welding can be accounted for in the solid-state welding process, where the intimate weld interface is formed under plastic deformation at elevated temperatures and surface diffusion which enables re-arrangement of the interface to perfectly fit. Thomas et al. [11] pointed out that the solid-state bonds in the interface can be accomplished at extremely concise times (~milliseconds). However, weld nuggets form under longer weld times (hundreds of milliseconds); thus, the resulting joints can be formed either through solid-state or fusion processes. In the research of Chun et al. [22], the inhomogeneous microstructures, including the unmixed Al-Si coating layer and the second-phase $Fe_3(Al, Si)$ intermetallic compound at the edge of the nugget, were responsible for the poor weld mechanical property. In Nielsen's model [25], friction was implemented between the square nut projections and the sheets during the welding process simulation, which increases the accuracy of electro-thermo-mechanical RPW modeling. Gintrowski et al. [26] reported that the RPW joint between aluminum and copper has good electrical and mechanical properties despite the brittle intermetallic AlCu layer in the interface. Saad [27] studied the effect of resistance spot welding parameters on copper and brass alloys, specifically analyzing the metallographic structures and joint properties, as well as fracture failure. However, there is limited information available on the formation mechanism of welding interfaces and their effect on joint performance. Therefore, the purpose of this research is to investigate the effect of resistance projection welding parameters on bronze/brass alloy joints and to explore the formation mechanism of the interface.

We weld phosphorus bronze and brass alloys using resistance projection welding technology in this work. Scanning electron microscopy (SEM), electron back-scattered diffraction (EBSD), and energy dispersive spectroscopy (EDS) were used to characterize the welded samples. This work aims to investigate the weld behavior and microstructure evolution of dissimilar copper alloy welded joints.

## 2. Materials and Methods

The base metals for the dissimilar welded joints are XYK-6 phosphorus bronze and H62 brass alloy sheets of 0.5 mm thickness. The chemical compositions of the base metals were determined by an inductively coupled plasma optical emission spectrometer (ICP-OES, Spectro Blue II, Germany), as shown in Table 1. The conductivity and mechanical properties of the base metals are presented in Table 2. It shows that the H62 brass has lower strength and conductivity compared to XYK-6 phosphorus bronze. Figure 1a shows that the melting point of H62 brass is about 900 °C, which is close to the boiling point temperature of zinc. Thus, the zinc in its structure is prone to volatilization when the brass is melted. The melting point of XYK-6 phosphorus bronze is approximately between 1060 °C and 1080 °C, which is higher than that of H62 brass (Figure 1b).

**Table 1.** Chemical composition of the materials.

| Materials | Chemical Composition % | | | | | | |
|---|---|---|---|---|---|---|---|
| | **Cu** | **Zn** | **Sn** | **P** | **Fe** | **Ni** | **Pb** |
| H62 | 60.5~63.5 | Bal. | - | - | ≤0.07 | - | ≤0.09 |
| XYK-6 | Bal. | 1.00 | 1.8~2.7 | 0.03~0.10 | 0.10 | 0.1~0.4 | 0.02 |

**Table 2.** Properties of the materials.

| Materials | Conductivity (%IACS) | Tensile Strength (MPa) |
|---|---|---|
| H62 | 25 | 350~470 |
| XYK-6 | 32 | 480~530 |

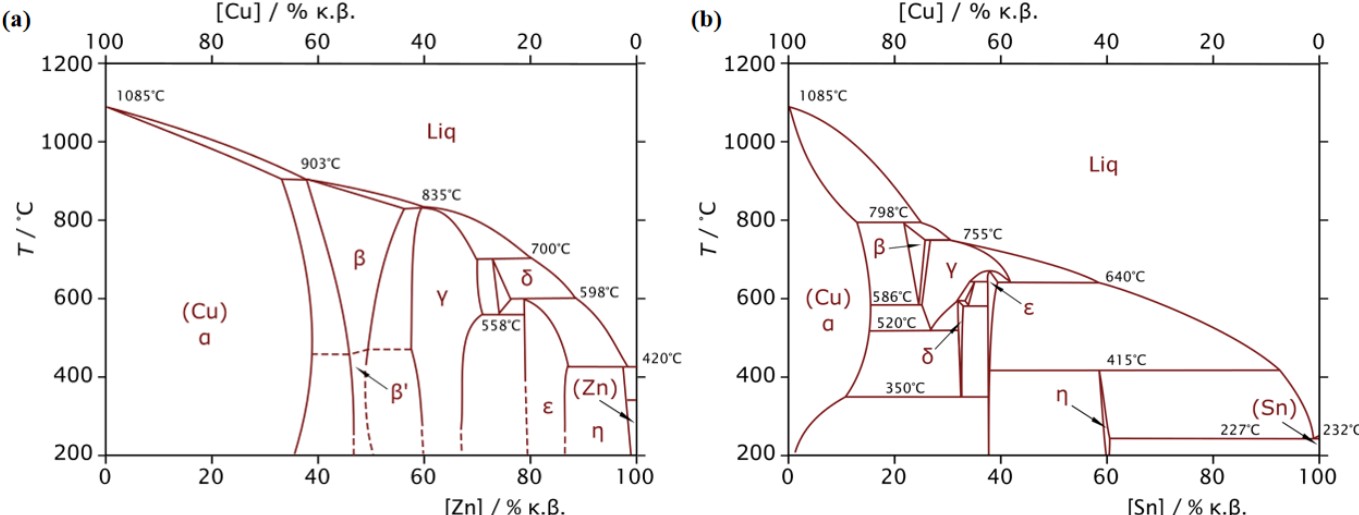

**Figure 1.** (**a**) Cu-Zn phase diagram and (**b**) Cu-Sn phase diagram [28].

The projection welded samples were produced using a 16 kVA mid-frequency inverter DC pedestal-type resistance-spot welding machine, as shown in Figure 2a. A high heat resistance W80Cu20 tungsten copper bar rod electrode with a flat surface diameter of 8 mm was used. Figure 2b,c shows the RPW illustration and the dimensions of the welded specimen, respectively. The projections are formed by stamping with self-made molds. Prior to welding, all coupons are cleaned by an ultrasonic cleaning machine with alcohol cleaner to avoid surface contamination. XYK-6 phosphorus bronze with an embossed projection was placed on a flat H62 brass in such a way that the projection side faced downwards as shown in Figure 2c. During the welding process, a welding current is locally applied to overlapping workpieces clamped together under pressure by the electrode.

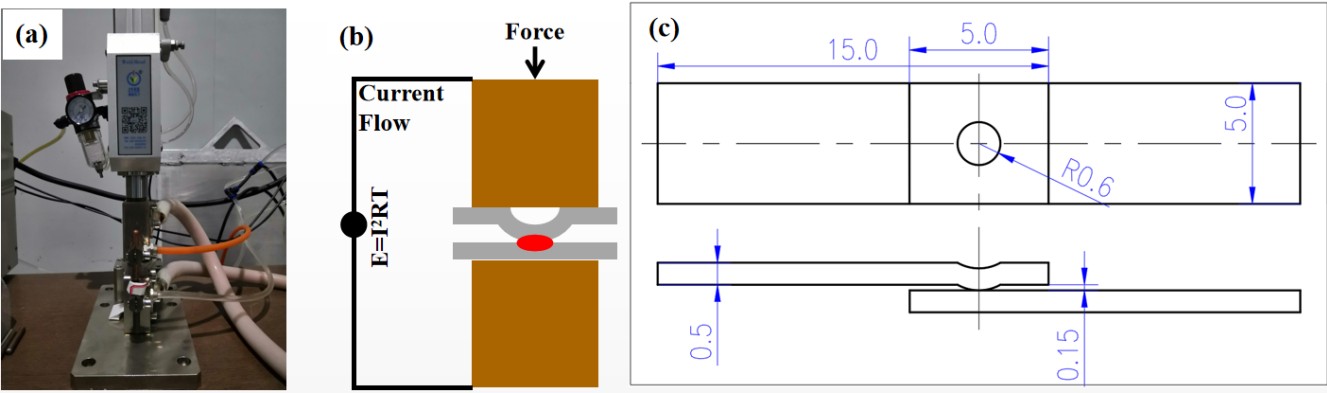

**Figure 2.** (**a**) Experimental equipment; schematic figures of (**b**) the projection weld illustration and (**c**) a welded specimen (all dimensions in mm).

The welding schedules are listed in Table 3. The mechanical property as the maximum tensile shear load and the electrical resistance of the RPW joints were thus determined for each specimen. The mechanical test was carried out as a tensile lap shear test (2 mm/min) on a Zwick/Roell Z010 (max. 10 kN) tensile testing machine. The electrical resistance tests were conducted using a double bridge circuit device at room temperature.

**Table 3.** Welding parameters used in the experiments.

| Electrode Force (N) | Welding Current (kA) | Welding Time (ms) | Cooling Time (ms) |
|---|---|---|---|
| 200 | 3.5, 4.5, 5.5, 6.5 | 20 | 10 |

After welding, the metallographic samples were cut into cross-sectioned joints by a wire electrical discharge machine. The samples were grounded using SiC paper with a grit size of 500, 1000, and 2000 in sequence, finally polished with 0.3 μm $\alpha$-$Al_2O_3$ and etched in aqueous solutions of $FeCl_3$/HCl for 10 s at room temperature. The microstructures of the weld joints were characterized by a Leica EC3 optical microscope and a field emission scanning electron microscope (FESEM, FEI Quanta 650, Hillsboro, OR, USA) with a 20 kV accelerating voltage. The elemental distribution in the weld joint was determined with energy dispersive spectroscopy (EDS, Oxford Instruments, Oxford, UK). The specimens for the EBSD measurements were prepared, mechanically grounded, polished, and finally polished with Vibratory Polisher (AZoNetwork UK Ltd., Manchester, UK) for 2 h. The EBSD measurements were conducted using a field-emission scanning electron microscope (EBSD, Oxford HKL Channel 5, Oxford Instruments, Oxford, UK) at an accelerating voltage, spot size, and step size of 20 kV, 4 μm, and 0.2 μm, respectively. The micro-hardness of the sample was determined using a Vickers hardness tester under 100 g load for 10 s, and the indentations were spaced 0.1 mm apart.

## 3. Results and Discussion

### 3.1. Properties of the Weld Joints

Figure 3 shows the tensile shear load and resistance of the brass/bronze (H62/XYK-6) RPW joints under different welding currents. In the welding current range from 3.5 to 5.5 kA, the tensile shear load of the joint increased with the increasing welding current. When the welding current was 5.5 kA, it reached the maximum value of 273.60 kN. When the welding current exceeded 5.5 kA, the load capacity of the RPW joints decreased slightly as the welding current increased. The resistance of the joint decreased sharply with the increasing welding current when the welding current ranged from 3.5 to 5.5 kA. As the welding current increases, the resistance of the joint drops rapidly. However, the joint resistance decreases much more slowly when the welding current exceeds 5.5 kA. This is because more heat is generated in the weld zone as the welding current increases, resulting in a larger weld bond area and therefore lower joint resistance.

### 3.2. Macro- and Micrographs of the Weld Joints

Figure 4a–d shows the transverse macro cross-section of an XYK-6 projection welded onto a flat H62 sample under the welding currents of 3.5 kA, 4.5 kA, 5.5 kA, and 6.5 kA, respectively. In the case where the current was low (3.5 kA), a small discontinuous gap was observed between the interfacial boundary (see Figure 5a). Due to the low melting point of brass and the high melting point of bronze, the brass first softens during the welding process, and the bumps of the bronze squeeze the brass material. As the welding current increases, the heat generation increases, and the area where the brass softens increases. The bronze bump extruded the brass. During the welding process, the bronze bump moves into the softer brass material without being deformed. When the welding current reached 5.5 kA, a thin layer of the solidified structure appeared at the bronze interface (as shown in Figure 5b). When the welding current reaches 6.5 kA, there is not only a solidified structure at the bronze interface, but also the excessive welding heat causes more melting of the brass, and the bronze bump squeezes the molten brass so that the partially molten brass is squeezed out for welding. At the interface, splashes are formed, and it can be seen that there are hole defects in the solidified brass structure, which is due to the volatilization of zinc in the molten brass structure. During the whole welding process, the bronze did not melt. When the welding current is small, the interface does not melt and weld, but it

does in solid-state diffusion welding. When the welding current increases, the interface is converted from solid-state diffusion welding to melting welding. Since zinc in brass are volatile, the welding current cannot be too large. Otherwise, the overheated welding heat will cause more melting of the brass, causing serious zinc volatilization and splashes, resulting in defective holes and weakening the joint connection performance.

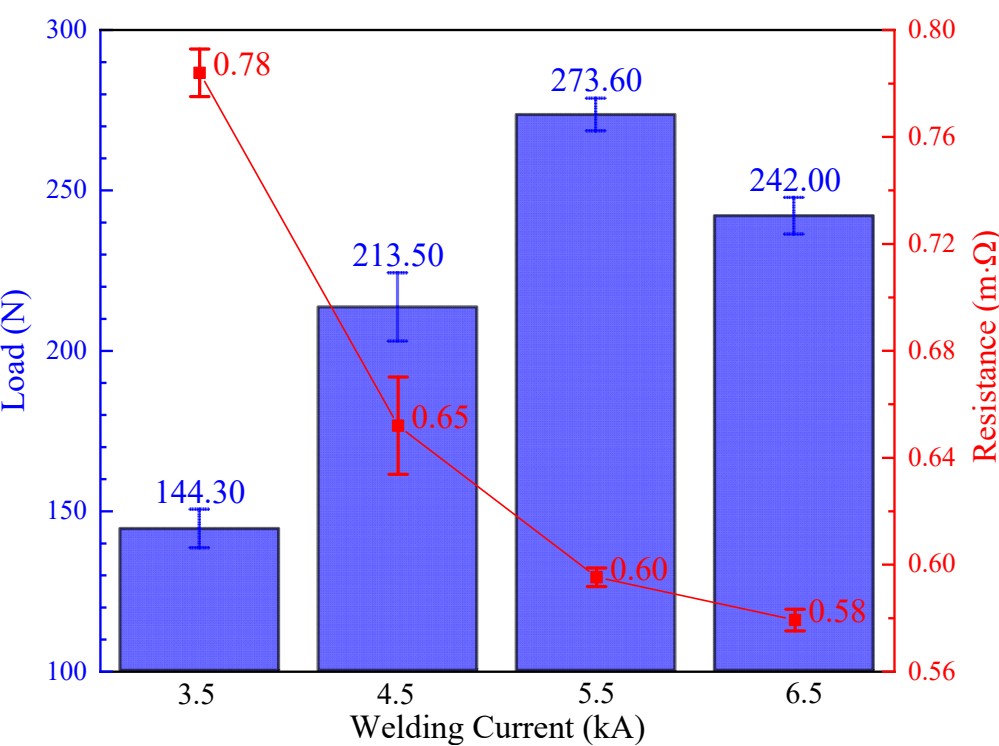

**Figure 3.** Effect of the welding current on the tensile shear load and resistance of the RPW joints.

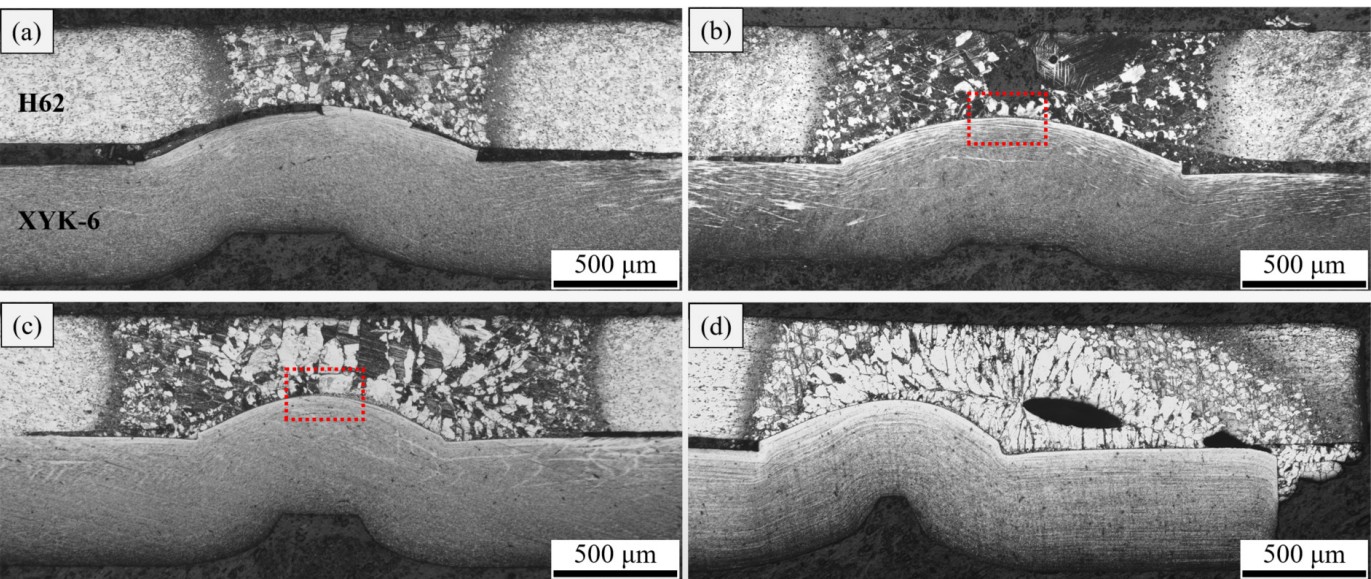

**Figure 4.** Traverse macro cross-section of an XYK-6/H62 projection weld: (**a**) 3.5 kA joint; (**b**) 4.5 kA joint; (**c**) 5.5 kA joint; (**d**) 6.5 kA joint.

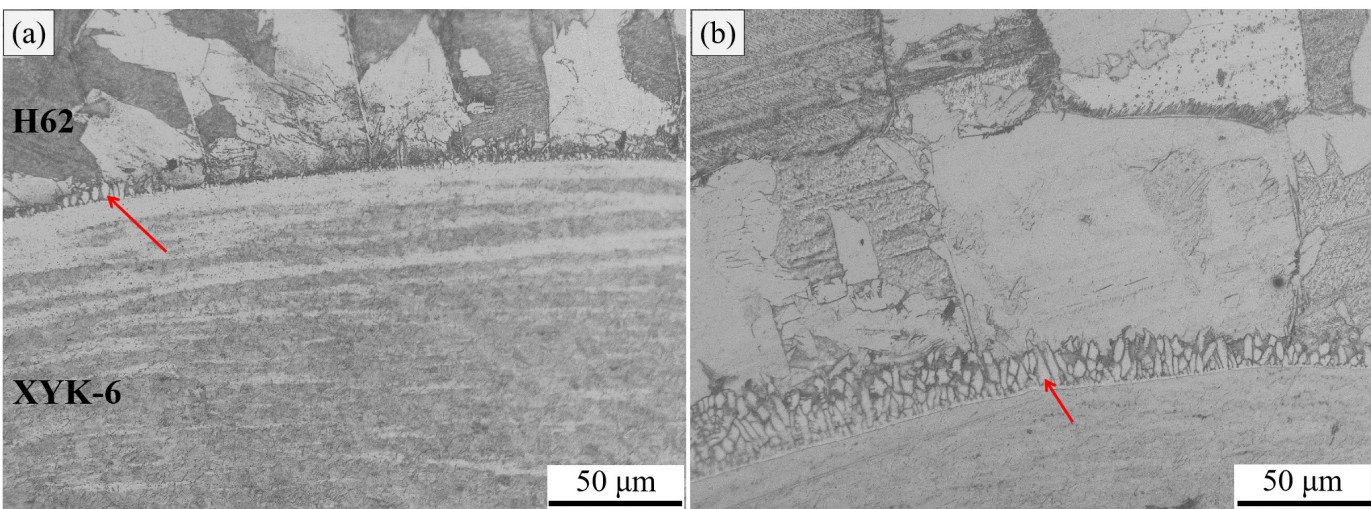

**Figure 5.** Optical micrograph of the H62/XYK-6 RPW interface (**a**,**b**) corresponding to the red dotted box area in Figure 4b and c, respectively.

Figure 5a shows that scattered and a few solidified columnar crystal structures are observed, indicating the onset of melting at local contact points during RPW at a welding current of 4.5 kA. Furthermore, as shown in Figure 5b, more solidified columnar crystal structures are obviously found and they are contiguous with each other, suggesting that the increasing heating accelerates the melting at the interface of H62/XYK-6 at a welding current of 5.5 kA. Therefore, a good-quality weld can be fabricated when the welding current is 5.5 kA. In the following experiments and the result analysis, these fixed parameters were chosen.

### 3.3. Metallurgical Characteristics

The EBSD results for H62 and XYK-6 base metals are depicted in Figures 6 and 7, respectively. The majority of grains in the H62 base metal were formed with a <111> crystal orientation, while those in the XYK-6 base metal had a prevalence of <101> crystal orientation. The average grain size in the H62 base metal and the XYK-6 base metal were 5.04 μm (Figure 6e) and 1.09 μm (Figure 7e), respectively. The fraction of high-angle grain boundaries (HAGBs), low-angle grain boundaries (LAGBs), and twin boundaries (TBs) in the H62 base metal were 77.7%, 21.6%, and 0.7% of the total grain boundaries (Figure 6f), respectively. In contrast, in the XYK-6 base metal, they were 66.9%, 32.6%, and 0.5% of the total grain boundaries (Figure 7f), respectively.

Additionally, the H62 base metal consisted of 5.2% recrystallized grains, 5.3% subgrains, and 89.5% deformed grains (Figure 6g), while the XYK-6 base metal consisted of 5.9% recrystallized grains, 12.4% subgrains, and 81.7% deformed grains (Figure 7g). In the kernel average misorientation (KAM) map of Figures 6d and 7d, the KAM values were macroscopically uniform but showed high values at the grain boundaries in both the H62 and XYK-6 base metals. Figures 6h and 7h showed that the local misorientation had a unimodal distribution with an average value of 1.80 and 1.39 in the H62 and XYK-6 base metals, respectively. The KAM value can be used to reflect the dislocation density [29], Thus, the higher KAM values observed at grain boundaries can be attributed to a higher density of geometrically necessary dislocations (GNDs) stored in the deformed structure. Based on the corresponding results shown in Figures 6c and 7c, it can be inferred that both of the H62 and XYK-6 base metals were subjected to deformation before use.

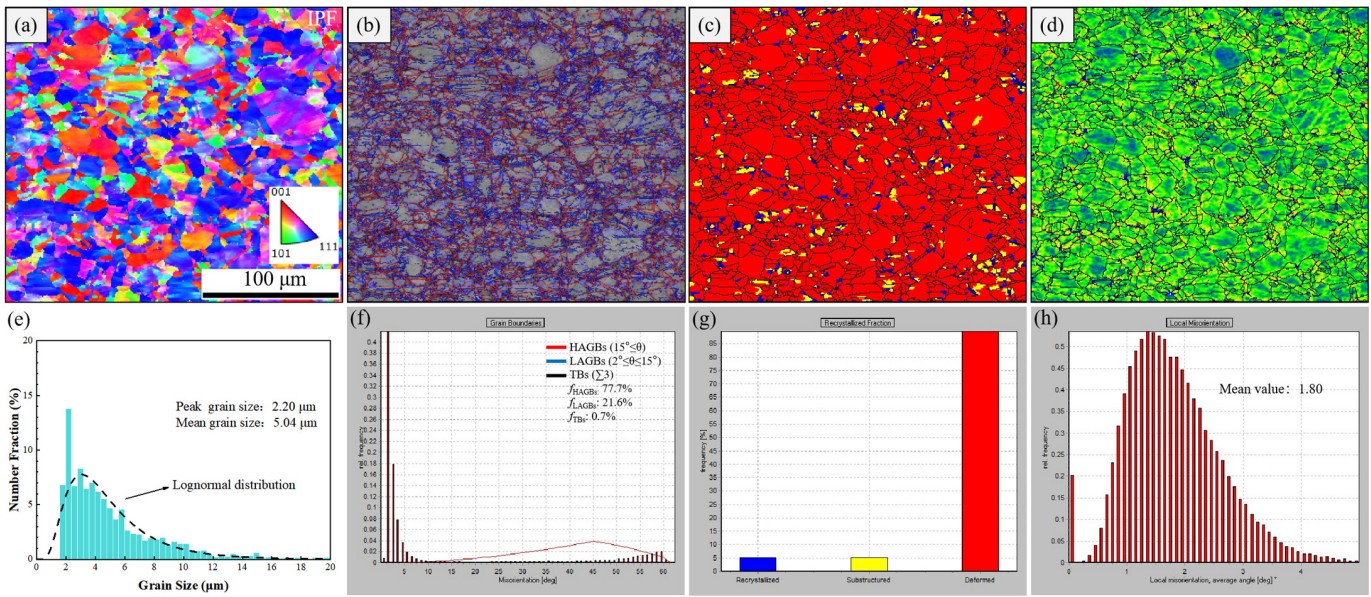

**Figure 6.** EBSD images of the H62 base metal: (**a**) the inverse pole figure maps; (**b**) GBs (high-angle grain boundaries (HAGBs with θ ≥ 15°)) are marked in red lines, low-angle grain boundaries (LAGBs with 2° < θ < 15°) are marked in blue lines, and twin boundaries (TBs) are marked in black lines; (**c**) recrystallized grains (in blue), subgrains (in yellow), and deformed grains (in red); (**d**) kernel average misorientation (KAM) map; (**e**) corresponding statistical distribution of grain size; (**f**) fraction of the grain boundaries; (**g**) fraction of the recrystallization; (**h**) local misorientation angle distribution.

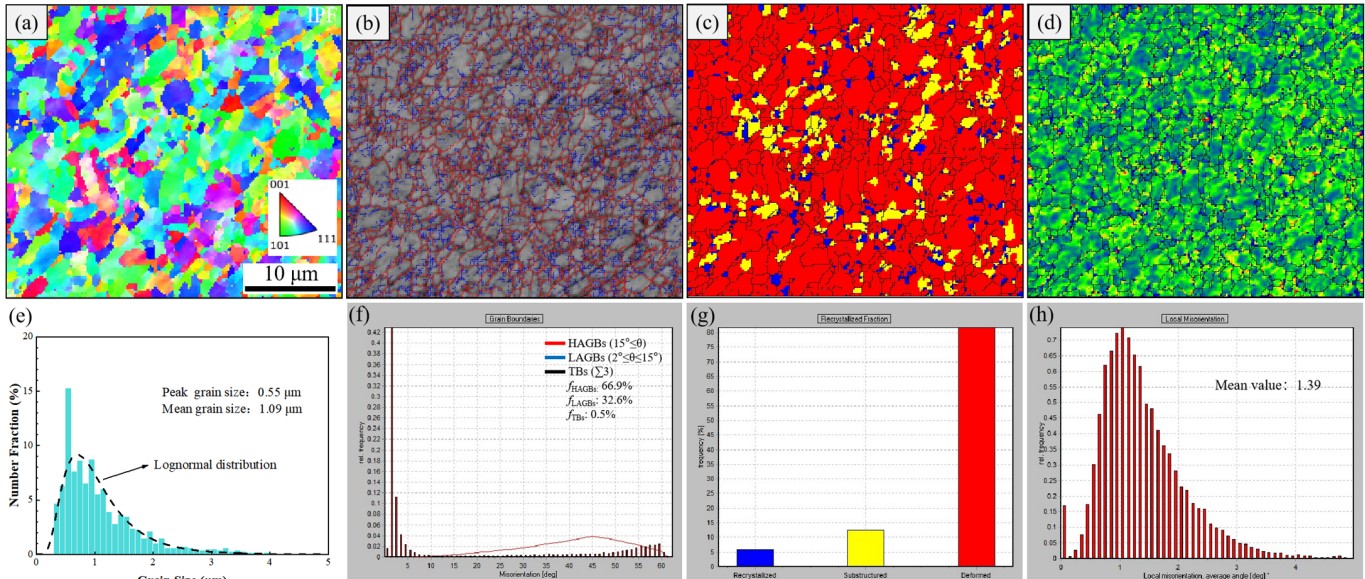

**Figure 7.** EBSD images of the XYK-6 base metal: (**a**) the inverse pole figure maps; (**b**) GBs (high-angle grain boundaries (HAGBs with θ ≥ 15°)) are marked in red lines, low-angle grain boundaries (LAGBs with 2° < θ < 15°) are marked in blue lines, and twin boundaries (TBs) are marked in black lines; (**c**) recrystallized grains (in blue), subgrains (in yellow), and deformed grains (in red); (**d**) kernel average misorientation (KAM) map; (**e**) corresponding statistical distribution of grain size; (**f**) fraction of the grain boundaries; (**g**) fraction of the recrystallization; (**h**) local misorientation angle distribution.

To reveal the interfacial bonding quality and mechanism, detailed microstructural characterization after welding was carried out on a typical H62/XYK-6 weld interface, as shown in Figure 8. It can be observed from Figure 8a that no significant orientation

prevailed at the weld bonding interface. The grains of brass grow significantly after welding (~100 μm) compared to the grain before welding (~5.04 μm). On the other hand, the growth of the bronze grains (black dashed box) during welding is relatively slow, from 1.09 μm (before welding) to 2.82 μm (after welding). The fraction of HAGBs, LAGBs, and TBs at the weld interface were 44.2%, 54.9%, and 0.9% of the total grain boundaries (Figure 8f), respectively. In addition, the H62/XYK-6 weld interface contains 21.8% recrystallized grains, 55.9% subgrains, and 22.3% deformed grains (Figure 8g). In the KAM map of Figure 8d, the KAM was macroscopically uniform. Figure 8h shows the local misorientation is unimodal distribution. The average value is 0.71 at the H62/XYK-6 weld interface, which is lower than the H62 and XYK-6 base metals. Due to heat generation and pressure during welding, dynamic recrystallization takes place at the welding interface. As a result, the proportion of the deformed structure decreases while the recrystallized proportion increases in comparison to the two alloys before welding. This is evident from Figures 6f and 7f. Furthermore, the KAM values after welding are reduced as shown in Figure 8h.

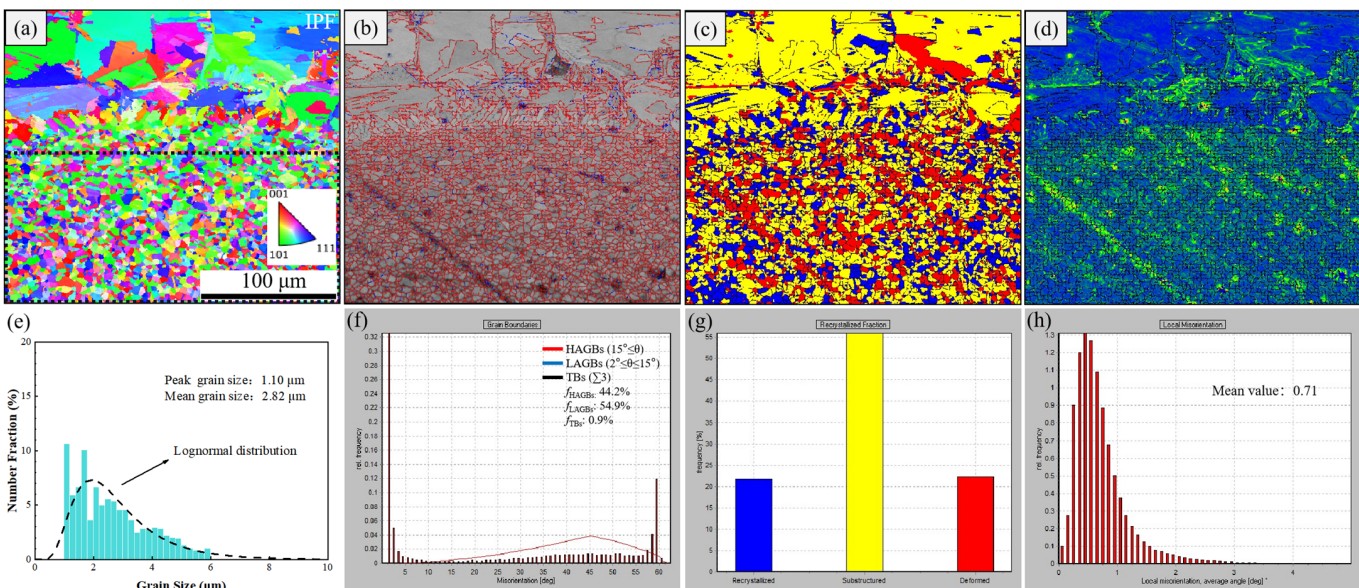

**Figure 8.** EBSD images of the H62/XYK-6 weld interface: (**a**) the inverse pole figure maps; (**b**) GBs (high-angle grain boundaries (HAGBs with θ ≥ 15°)) are marked in red lines, low-angle grain boundaries (LAGBs with 2° < θ < 15°) are marked in blue lines, and twin boundaries (TBs) are marked in black lines; (**c**) recrystallized grains (in blue), subgrains (in yellow), and deformed grains (in red); (**d**) kernel average misorientation (KAM) map; (**e**) corresponding statistical distribution of grain size; (**f**) fraction of the grain boundaries; (**g**) fraction of the recrystallization; (**h**) local misorientation angle distribution.

In order to better understand the formation mechanism of the interface, the elemental distributions of Cu, Zn, and Sn at the interface of two alloys after RPW processing are shown in Figure 9. It can be seen from Figure 9a that Cu is slightly rich at the bottom, and Zn is obviously rich at the top of the observed area, indicating that the composition of the columnar crystals is completely consistent with that of brass. It can be verified that the columnar crystals are all brass structures. There is no clear boundary for Sn in the elemental surface scan, possibly because its content is already low. EDS line scan analysis was performed at the welding interface to determine the elemental distribution. As shown in Figure 9b, the yellow line in the figure represents the EDS line scan position. The EDS line scan shows an elemental transition diffusion layer of about 1.5 μm between the brass columnar crystal and XYK-6. According to the metallographic structures, as shown in Figure 9, it can be inferred that the welding temperature is just over the melting point of H62 brass but never exceeded the melting point of XYK-6. As a result, the interfacial

reaction in this area is mostly between liquid H62 and solid XYK-6, generating a reaction layer consisting of a columnar crystal solidification layer and a diffusion layer.

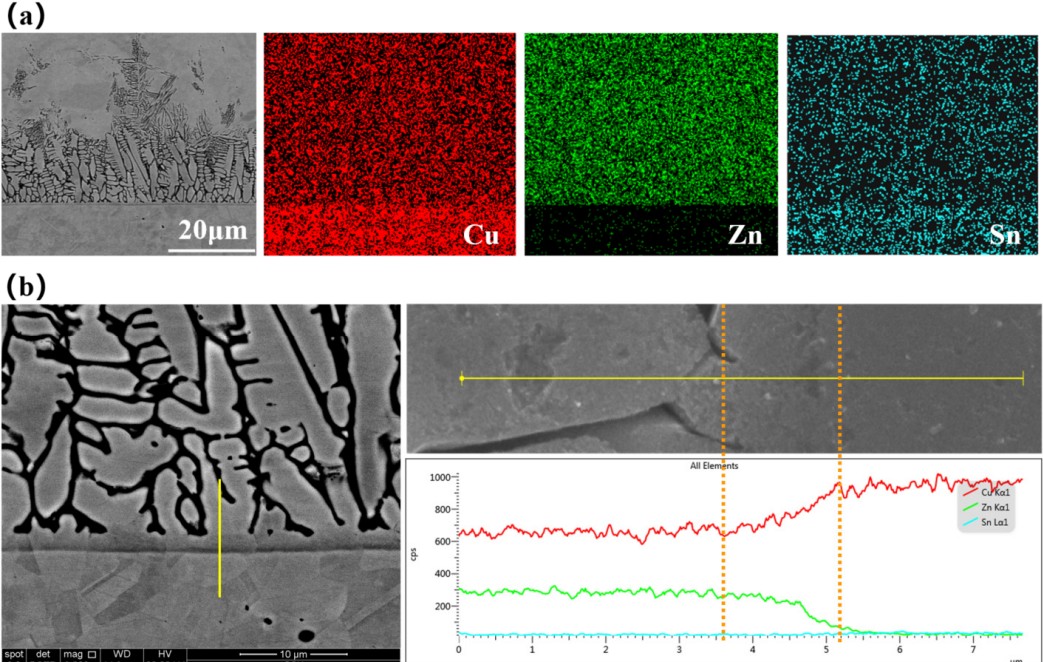

**Figure 9.** Elemental distribution of Cu, Zn, and Sn at the H62/XYK-6 RPW interface. (**a**) EDS mapping and (**b**) EDS line path on the SEM image and the curves of the EDS elemental analysis.

Since only a few results have been found in the literature on the formation of a diffusion layer at the surrounding area of the brass–bronze interface by RSW/RPW, we propose a hypothesis here of a three-stage reaction mechanism between near solid/solid brass and solid bronze as illustrated in Figure 10b–d. Figure 10a depicts the initial squeezing stage before welding. High compression stresses the electrode applied and arouses high strain at the H62/XYK-6 interface, producing close contact regions. When the electric current passes through the contact regions and generates Joule heat (Figure 10b), the H62 materials were softened firstly by the elevated temperature, and the XYK-6 bump extruded softened H62. Thus, the contact regions increased with diminishing gaps. Furthermore, as shown in Figure 10c, heating accelerates the softened H62 materials which fabricate the joining region by atomic interdiffusion, leading to solid-phase diffusion welding between the solid softened H62 and solid XYK-6 alloys. Additionally, as the welding current is increased, local melting occurs at the contact interface of the brass due to increased heat generation. This leads to liquid-phase diffusion between the liquid H62 brass and the solid XYK-6. The molten brass takes the interfacial diffusion layer as its crystalline surface and solidifies in the cooling direction to form dendrites (Figure 10d), thus achieving a strong welded joint.

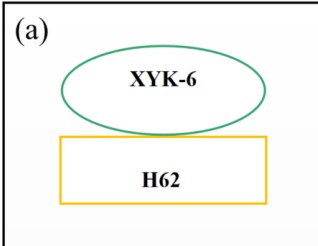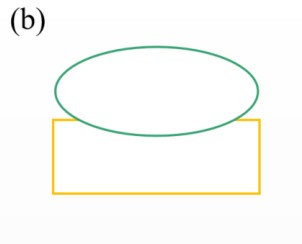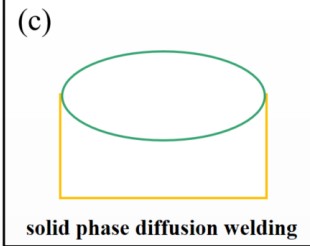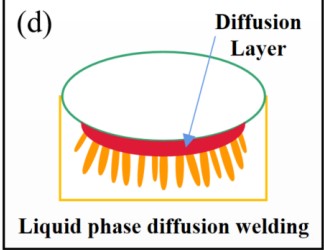

**Figure 10.** Schematics of the H62/XYK-6 interface at different stages during RPW: (**a**) the initial squeezing stage and (**b**–**d**) the welding stage.

Figure 11 shows the photograph of the fractured specimen of the RPW joint at a welding current of 5.5 kA. It can be seen from Figure 11 that the fracture occurred on the metal side of XYK-6 instead of the welding interface, resulting in the intactness of the welding interface. The schematic of the fracture mechanism is shown in Figure 11b. The interface strength is high after welding due to good interface bonding. Hence, during the tensile loading process, cracks can only develop from the interface under stress resulting in the fracture passing through the convex point between the two pieces, while the welding interface remains intact. It can be inferred that when the welding current is low, no effective bonding is formed at the interface, and its fracture will occur at the welding bump interface without passing through the bump.

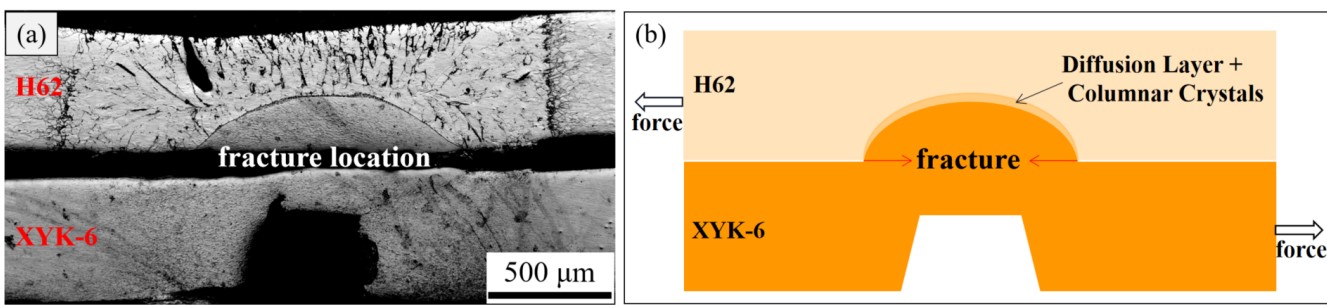

**Figure 11.** (**a**) Photograph of the fractured specimen and (**b**) schematic of the fracture.

In summary, the welding current has an evident effect on the interface layer. When the welding current is low, the solidified columnar crystal structures at the interface layer are fewer and discontinuous, resulting in low interface bonding strength. This is because interface bonding relies mainly on solid-phase diffusion. However, increasing the welding current leads to increased heat input, which generates more molten brass at the interface and results in the formation of continuous solidified columnar crystal structures in the interface layer. This transition from solid-phase diffusion welding to liquid-phase diffusion welding improves the interface bonding strength.

## 4. Conclusions

The microstructure and performance of an XYK-6 phosphorus bronze/H62 brass joint produced by resistance projection welding were investigated. The main conclusions are summarized as follows:

1. Resistance projection welding is a suitable method for producing robust welds between bronze and brass. The welded joints show a good quality tensile shear load and electrical conductivity.
2. The tensile shear load of the joint increased as the welding current increased. When the welding current was 5.5 kA, it reached the maximum value of 273.60 kN. The resistance of the joint decreased with the increase in the welding current.
3. At a welding current of 5.5 kA, a reaction layer is formed at the interface adjacent to the boundary of phosphorus bronze. This layer consists of a columnar crystal solidification layer and a diffusion layer. A tensile shear fracture occurred on the metal side of XYK-6, passing through the bump instead of the welding interface.
4. The formation mechanism of the welding interface is revealed for the first time, demonstrating a shift from solid-phase diffusion welding to liquid-phase diffusion welding, ultimately resulting in a robust welded joint. The results revealed that resistance projection welding is an effective method for welding copper alloys, suggesting the broad prospects of this technology in welding electrical parts, such as in the electrical and automotive industries.

**Author Contributions:** R.L. was the principal investigator of the research and wrote the original draft. X.Z. and T.L. carried out the welding tests. X.S. and P.C. analyzed the microstructure of the welded samples. Q.L. and X.Y. completed the review and editing. W.Z., Y.C. and Y.L. provided supervision. All authors have read and agreed to the published version of the manuscript.

**Funding:** This research received no external funding.

**Institutional Review Board Statement:** Not applicable.

**Informed Consent Statement:** Not applicable.

**Data Availability Statement:** Not applicable.

**Conflicts of Interest:** The authors declare no conflict of interest.

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
