# Peer review of "Microstructure and Properties of Phosphorus Bronze/Brass Joints Produced by Resistance Projection Welding"

_coatings, doi:10.3390/coatings13061032_

Round 1

Reviewer 1 Report

Review report: Microstructure and properties of phosphorus bronze/brass joints produced by resistance projection welding. Work is presented well with good publishing quality and can be accepted after the following corrections:  

1.       Add some quantitative results at the end of the abstract section.

2.       In place of citing multiple references, explain the individual work of the author and try to make a bridge between current and previous work.

3.       Novelty and application: Add a separate section for novelty and application of work.

4.       Materials and methods: Section is presented well but need some corrections. Add a detail of experimental set up instead of a schematic image. Also add the parameters and provide the mechanical properties of the used material.

5.       How was the composition of base metal analysed? If it was referred from some other publication, Add proper reference.

6.       Discuss about the mechanical properties of the base plates.

7.       Add the image of the plate after welding and their micrograph.

8.       Add discussion related to the welding parameters.

9.       Add standard used for specimen preparation. Also provide complete detail about the tensile shear load. Image of the specimen after testing their fracture location.

10.    Explain the mechanism related to the effect of current on tensile strength.

11.    Add detail discussion on interface characterization.

NA

Author Response

Response to Reviewers

Dear Editor:

We would like to thank you for your kind suggestions on our manuscript entitled “Microstructure and properties of phosphorus bronze/brass joints produced by resistance projection welding” (Manuscript ID: coatings-2428703) submitted for publication in Coatings. Meanwhile, we appreciate very much for the kindness of the reviewers, who gave us so many valuable comments and sound suggestions to the improvement of our manuscript. The revisions have been carefully carried out according to the reviewers’ and your comments. The revised parts are highlighted in yellow background in the revised manuscript. Reviewer’s comments and the corresponding revisions are listed as follows:

Reviewer: 1

  1. Add some quantitative results at the end of the abstract section.

Response: Thanks for your constructive suggestions, we have added some quantitative results at the end of the abstract sections in the modified manuscript (line 20-23).

  1. In place of citing multiple references, explain the individual work of the author and try to make a bridge between current and previous work.

Response: Thanks for your constructive suggestions, we have corrected it in the modified manuscript (line 61-69).

  1. Novelty and application: Add a separate section for novelty and application of work.

Response: Thanks for your constructive suggestions, we have added the novelty and application of work in the modified manuscript (line 311-315).

  1. Materials and methods: Section is presented well but need some corrections. Add a detail of experimental set up instead of a schematic image. Also add the parameters and provide the mechanical properties of the used material.

Response: Thanks for your constructive suggestions, we have added a detail of experimental setup in the modified manuscript (line 94-99). The welding parameters are provide in Table 3 and the mechanical properties of the used material are presented in Table 2.

  1. How was the composition of base metal analysed? If it was referred from some other publication, Add proper reference.

Response: Thanks for your suggestions. In fact, the chemical composition of base metals was determined by an Inductively Coupled Plasma Optical Emission Spectrometer (ICP-OES, Spectro Blue â…¡, Germany). And we have added the relative description about the composition testing technique in the modified manuscript (line 77-81).

  1. Discuss about the mechanical properties of the base plates.

Response: Thanks for your suggestions, we have added discuss about the mechanical properties of the base plates in the modified manuscript (line 80-81)

  1. Add the image of the plate after welding and their micrograph.

Response: Thanks for your suggestions. The plate after welding is too small, and the image could not present valuable information for this article, so it was not added. The micrographs are provide in Figure 4, 5, 6, 7, 8, 9.

  1. Add discussion related to the welding parameters.

Response: Thanks for your suggestions, we have added the discussion related to the welding parameters in the modified manuscript (line 281, line 301-309).

  1. Add standard used for specimen preparation. Also provide complete detail about the tensile shear load. Image of the specimen after testing their fracture location.

Response: Thanks for your constructive suggestions. The specimen preparation has no standard and which is designed according to the research needs. The complete detail about the tensile shear and the image of the specimen after testing their fracture location is added in Figure11, and we have added the relative description in the modified manuscript (line 277-286).

  1. Explain the mechanism related to the effect of current on tensile strength.

Response: Thanks for your constructive suggestions, we have added the explain about the mechanism related to the effect of current on tensile strength in the modified manuscript (line 284-286, line 293-296).

  1. Add detail discussion on interface characterization.

Response: Thanks for your constructive suggestions, we have added detail discussion on interface characterization in the modified manuscript (line 249-256).

In summary, the whole manuscript has been revised carefully according to the reviewers’ comments and suggestions. We would be very grateful if the revised manuscript could be finally accepted for publication in Coating. Thanks for your consideration in advance and we are looking forward to hearing from you at your earliest convenience.

With best regards!

Yours sincerely,

Ruilin Lai, PhD

Central South University, China

Reviewer 2 Report

The article titled "Microstructure and Properties of Phosphorus Bronze/Brass Joints Produced by Resistance Projection Welding" presents a study on the fabrication of lap joints between embossed projection phosphorus bronze and flat brass using resistance projection welding (RPW). The main objectives of the study were to investigate the microstructure and properties of the welded joints and evaluate the effectiveness of RPW for welding copper alloys. Experimental results showed that the bronze projection moves into the softer brass without deformation during the welding process. The tensile shear loads of the joints reached a maximum value of 273.6 N at a welding current of 5.5 kA. A reaction layer consisting of a columnar crystal solidification layer and a diffusion layer was formed at the interface beside the bronze boundary. The findings indicated that resistance projection welding is an effective method for welding copper alloys, suggesting its potential in welding electrical parts. The base metals used for the welded joints were phosphorus bronze and brass alloys. The melting points of the materials were considered to ensure proper welding conditions. The welding process was performed using a resistance-spot welding machine with specific parameters. The mechanical properties, such as tensile shear load, and electrical resistance of the RPW joints were determined for each specimen. The results showed that increasing the welding current led to higher tensile shear loads and lower joint resistance. However, excessive welding heat at higher currents caused defects and weakened the joint connection performance. Microstructural analysis revealed the formation of solidified structures and the occurrence of melting at the interface between the bronze and brass. The welding current of 5.5 kA was identified as the optimal condition for producing good-quality welds. The microstructure of the base metals, including grain orientation and boundaries, was also analyzed.  In conclusion, the study demonstrated that resistance projection welding can effectively join phosphorus bronze and brass, providing promising prospects for its application in welding electrical parts. The findings contribute to the understanding of microstructure-property relationships in welded joints and offer valuable insights for optimizing the welding process.

In my opinion, the work is well-written, well-structured, and adequately addresses all the proposed questions. Therefore, it is a pleasure to recommend it for publication.

Author Response

Reviewer: 2

In my opinion, the work is well-written, well-structured, and adequately addresses all the proposed questions. Therefore, it is a pleasure to recommend it for publication.

Response: Thank you very much for your positive comments.

Reviewer 3 Report

Ruilin Lai et al reported a research paper titled "Microstructure and properties of phosphorus bronze/brass joints produced by resistance projection welding". Authors welded bronze/brass joints using resistance projection welding (RPW). Characterized the welded samples using SEM, EDS, and EBSD. Authors proposed a three stage reaction mechanism in the formation of diffusion layer at the brass-bronze interface during the welding process with the current of 5.5 kA. Overall the article is well written and fits the scope of the journal "Coatings".

Comments:

Figure 6,7,8 (f), (g), (h): title bars as well as X and Y titles and numbers are difficulty to see. Reviewer suggest authors increase the size of the letters and numbers so that audience can see the results from the figures easily.

Author Response

Reviewer: 3

  1. Figure 6,7,8 (f), (g), (h): title bars as well as X and Y titles and numbers are difficulty to see. Reviewer suggest authors increase the size of the letters and numbers so that audience can see the results from the figures easily.

Response: Thanks for your constructive suggestions, which help us improve our manuscript greatly. We have increased the resolution of the Figure 6, 7, 8 in the revised manuscript.

Round 2

Reviewer 1 Report

Accepted.

NA